# Species-specific detection of *Schistosoma japonicum* using the 'SNAILS' DNA-based biosensor
Alexander J. Webb [1], Qin-Ping Zhao[2], Fiona Allan[3,4], Richard J. R. Kelwick [1], Aidan M. Emery [3] & Paul S. Freemont [1,5,6] ✉

The neglected tropical disease schistosomiasis continues to be a global health concern, especially in low- and middle-income countries, with at least 250 million people infected worldwide and a further 779 million at risk of infection. *Schistosoma japonicum*, which is found in parts of South Asia, causes intestinal schistosomiasis in humans, as well as infecting up to forty other mammalian species. Therefore, novel diagnostics that can detect *S. japonicum* are desirable. In this study, we have further developed and refined the 'SNAILS' DNA-based biosensor technology for the detection of schistosomes and have applied this to the specific detection of *S. japonicum*. Of the four new 'SNAILS' probe pairs developed, one of our optimised designs successfully detected and differentiated between genomic DNA isolated from 16 *S. japonicum* cercariae isolated from sites in the People's Republic of China and laboratory-derived *Schistosoma mansoni* cercariae.

The debilitating neglected tropical disease (NTD) schistosomiasis, caused by parasitic trematode flatworms of the genus *Schistosoma*, continues to be a cause of concern to global health, especially in low- and middle-income (LMIC) countries[1]. Over 250 million people are infected worldwide, with a further 779 million at risk of infection[1–3]. Furthermore, the global burden of schistosomiasis in 2021 was estimated to be 2.19 million disability-adjusted life-years (DALYs)[4]. 'Schistosomiasis' is somewhat of a blanket term, covering a group of related but distinct diseases caused by several different species of schistosome – not all of which give rise to human disease. Even so, it has been noted that for parasitic diseases, schistosomiasis has the second highest mortality in humans, only behind that conferred by malaria[5]. Schistosomes are found in Africa, the Arabian Peninsula, South America, the People's Republic of China (PRC), the Philippines, Indonesia and more recently Corsica in Southern Europe[1,6]. The most prevalent agents of human schistosomiasis are *Schistosoma mansoni*, *Schistosoma haematobium*, *Schistosoma japonicum*, with an additional three species (*Schistosoma intercalatum*, *Schistosoma guineensis* and *Schistosoma mekongi*) each causing disease over a much more limited geographic range[1].

*S. japonicum*, which like *S. mansoni* causes intestinal schistosomiasis[1], is a zoonotic species that has more than 40 different domestic and wild mammal species able to act as definitive hosts[7,8]. The distribution of *S.*

*japonicum* is mainly localised to PRC, the Philippines and Indonesia, having been eliminated from Japan in the mid 1990's[1]. In similarity with other schistosome species, *S. japonicum* has a life cycle that alternates between asexual stages in a snail intermediate host and sexual reproduction in a mammalian definitive host. Within infected definitive hosts, such as humans, *S. japonicum* adult worms release eggs which are excreted from the host faecally. Upon contact with freshwater they hatch into free-swimming miracidia. These larvae then seek out and infect freshwater snails of the genus *Oncomelania*, inside of which they then develop into sporocysts, which then produce subsequent daughter sporocysts via asexual reproduction. These daughter sporocysts then produce numerous amounts of cercariae asexually. These cercariae, which are also a free-swimming stage, are shed from the freshwater snails and seek out the definitive host. Upon contacting the definitive host, the cercariae then penetrate the skin, loose their tails and become schistosomula. The schistosomula then migrate through the circulatory system to the mesenteric venules surrounding the large intestine where they mature into separate-sex adults. Mating pairs of adult worms then reproduce sexually, release eggs and the cycle repeats[1].

The disease schistosomiasis japonica has higher morbidity than schistosomiasis caused by other species, due to the higher number of eggs released by *S. japonicum*[1,9]. Therefore, as for other schistosomes of human and

[1]Section of Structural and Synthetic Biology, Department of Infectious Disease, Imperial College London, London, UK. [2]Department of Parasitology, School of Basic Medical Sciences, Wuhan University, Wuhan, Hubei Province, People's Republic of China. [3]Natural History Museum, London, UK. [4]London School of Hygiene and Tropical Medicine, London, UK. [5]The London Biofoundry, Imperial College Translation and Innovation Hub, White City Campus, London, UK. [6]UK Dementia Research Institute Care Research and Technology Centre, Imperial College London, Hammersmith Campus, London, UK. ✉e-mail: p.freemont@imperial.ac.uk

veterinary importance it is desirable for public health reasons to break the infective cycle. For example, since the 1950's PRC has used snail control and mass drug administration (MDA) programmes to break the transmission cycle[10]. Similar strategies have also been used to eliminate the disease from Japan[1,11]. Other methods have also become popular, such as water, sanitation and hygiene (WASH) interventions, which also include behavioural change components[12]. An important element of controlling schistosomiasis is the ability to detect for the parasites in patient samples (human and veterinary), in populations of *Oncomelania* snails, in other definitive hosts such as cattle, as well as in the environment such as bodies of fresh water. Therefore, it is essential to detect these parasites rapidly and specifically. Previously, we developed and validated a species-specific DNA-based biosensor for detecting and differentiating *S. mansoni* and *S. haematobium*, which we termed: **Sp**ecific **N**ucleic **Ac**Id **L**igation for the detection of **S**chistosomes or 'SNAILS'[13]. Here, we report on the successful application of the 'SNAILS' biosensor technology for the species-specific detection of *S. japonicum*.

## Results

### Identification of specific targets for the detection of *S. japonicum*

Our 'SNAILS' biosensor system was developed to detect and differentiate *S. mansoni* and *S. haematobium*[13] – two of the main species responsible for human schistosomiasis[1,13]. The 'SNAILS' DNA-based biosensor, which is designed to bind to single stranded 22-base long targets, is adapted from a light-up 'Spinach' aptamer-based biosensor employed to detect cancer cell-derived micro RNAs (miRNA)[14] and, more recently, to detect SARS-CoV-2 in clinical samples[15]. The 'SNAILS' biosensor technology comprises two probes (A and B), which recognise a different half (11 bases) of the respective 22-base target. The 5' end of probe A, which encompasses the 11-base target-complementary region, is phosphorylated, whilst probe B's 11-base target-complementary region comprises the 3' hydroxylated (-OH) end (Supplementary Table 1). If the two probes recognise and anneal to a 22-base single stranded DNA (ssDNA) target, then addition of T4 DNA ligase will catalyse the formation of a covalent bond between the 5' and 3' ends of the probes. Probe A also encodes a 3' T7 promoter while probe B also encodes the 'Spinach' aptamer. Therefore, after successful ligation and the addition of both a complementary T7 promoter oligonucleotide and T7 RNA polymerase, transcription of the 'Spinach' aptamer will occur. Subsequent binding of the transcribed aptamer with the DFHBI-1T fluorogen will then enable real-time fluorescence measurements[13]. In order to increase the specificity of the biosensor the probes were designed to have maximum target sequence divergence around the ligation junction of the target complementary region. Based on our previous study this also helps to ensure biosensor specificity.

As part of a previous *S. haematobium* – *S. mansoni* proof-of-concept study, we targeted the cytochrome c oxidase subunit 1 (*cox*1) mitochondrial gene and, by aligning a range of species-specific *Schistosoma cox*1 sequences (see methods) using MUSCLE[16], we were able to identify multiple 22-base long species-specific *Schistosoma cox*1 targets. In this study, we decided to further develop the 'SNAILS' biosensor technology and analyse its ability to detect for *S. japonicum* – another species responsible for human schistosomiasis[1] (Fig. 1a). A selection of *S. japonicum*-specific target regions previously identified (unpublished) were further validated by a second MUSCLE[16] alignment (Supplementary Fig. 1) of EU325878 and other *S. japonicum cox*1 nucleotide sequences obtained from GenBank (Supplementary Table 2). These included sequences obtained from a range of locations in PRC, the Philippines, Japan and Taiwan, thereby enabling the identification of *S. japonicum* targets covering a range of geographical strains/ isolates of this species. Four of these targets were selected for further molecular analysis.

### Validation of S. japonicum-specific probes

Probes 2, 3, 4 and 5 were subsequently designed and tested for their ability to detect the four *S. japonicum*-specific targets identified in this study (Fig. 1). Each probe was tested against its specific *S. japonicum* target (SJ_WT), as well as the corresponding target regions of *S. mekongi* (Smek_WT),

*Schistosoma malayensis* (Smal_WT), *S. mansoni* (SM_WT), *Schistosoma rodhaini* (SR_WT), *S. haematobium* (SH_WT), *Schistosoma bovis* (SB_WT), *S. guineensis* (SG_WT) and *Schistosoma curassoni* (SC_WT). Sequence variations for the corresponding target regions of these species, compared to that of *S. japonicum*, are indicated by red letters and the ligation junctions are underlined (Fig. 1b, c, f, g). In our previous proof-of-concept study, we determined that 50 nM of each probe half and 50 nM of each 22-base long synthetic target were appropriate for testing[13]. Probe 2 (Fig. 1b, d; Supplementary Fig. 2) and probe 5 (Fig. 1g, i; Supplementary Fig. 5) were both specific for their corresponding *S. japonicum* targets, with no off-target response to the corresponding target regions of the other *Schistosoma* species tested. Probe 4 was also able to detect for *S. japonicum* although there was a very slight off-target detection of the corresponding *S. mekongi* target sequence (Fig. 1f, h; Supplementary Fig. 4). Probe 3 was the least specific of the four probe designs, in that although it could detect *S. japonicum* it also had off-target responses to the *S. malayensis*, *S. mansoni* and *S. rodhaini* corresponding target sequences (Fig. 1c, e; Supplementary Fig. 3).

### Biosensor detection of purified PCR-derived target ssDNA

As for our previous 'SNAILS' proof-of-concept study[13], this current study requires a PCR amplification step followed by a Lambda exonuclease enzymatic treatment step to produce a pool of target ssDNA. Previously, we determined that 30 ng of the purified ssDNA target in the final 10 μL ligation reaction of the biosensor assay was appropriate for testing against 50 nM of each probe half[13]. In this study, we decided to amplify a 446-base fragment which encompasses the four target regions (Supplementary Fig. 1). To analyse the ability of the four probes to recognise and anneal to the PCR-derived 446-base long ssDNA samples, a *S. japonicum* control DNA sample was required. Therefore, the 446-base long gene fragment, corresponding to the sequence of EU325878, was synthesised and subsequently cloned into pCR-blunt-II-TOPO (pAJW333; Supplementary Table 3). Three separate PCR amplification reactions of plasmid pAJW333 were undertaken (SJ1-SJ3; Fig. 2), with the phosphorylated 3' primer p-3-SJ-cox1 (AJW1063; Supplementary Table 1), which enabled the subsequent removal of the 3' DNA strand via Lambda exonuclease treatment. As previously[13], 30 ng of ssDNA target was tested against 50 nM of each half probe. Both probe 2 and probe 5 were able to detect the three PCR-derived ssDNA targets, although probe 2 had a greater ability to anneal and detect based on the higher fluorescence output (Fig. 2; Supplementary Fig. 6). Whilst probes 3 and 4 were unable to anneal to and detect the three PCR-derived ssDNA targets (Fig. 2; Supplementary Fig. 6).

### S. japonicum-specific probe 2 detection of ssDNA targets from cercaria

Since probe 2 had the best ability to anneal to and detect the *S. japonicum* ssDNA obtained from the plasmid control samples, we wanted to test its ability to detect ssDNA derived specifically from *S. japonicum* biological samples. To this end, sixteen *S. japonicum* cercarial samples, which had previously been collected from a range of sites in PRC (Supplementary Table 6), as well as three laboratory-derived *S. mansoni* cercariae samples, were obtained from the Schistosomiasis collection at the Natural History Museum in London, UK (SCAN)[17]. Genomic DNA (gDNA) was released from punch disks as described within the material and methods section. The 446-base target regions were PCR amplified with primers 5-SJ-cox1/3-SJ-cox1 (*S. japonicum*-specific; AJW1061/AJW1062; Supplementary Table 1) or 5-SM-cox1/3-SM-cox1 (*S. mansoni*-specific; AJW1064/AJW1065. Supplementary Table 1), the resultant PCR products were gel purified and subsequently sequence verified. The obtained sequences (SJ1-SJ16 and SM1-SM3) are listed in Supplementary Table 6, with the *S. japonicum* sequences being deposited in the European Nucleotide Archive under the accessions OZ203289-OZ203304. To obtain ssDNA, gDNA samples SJ1-SJ16 and SM1-SM3 were PCR amplified using primers 5-SJ-cox1/p-3-SJ-cox1 (AJW1061/AJW1063; Supplementary Table 1) and 5-SM-cox1/p-3-SM-cox1 (AJW1064/AJW1066; Supplementary Table 1) respectively and Lambda exonuclease treated as detailed within the materials and methods

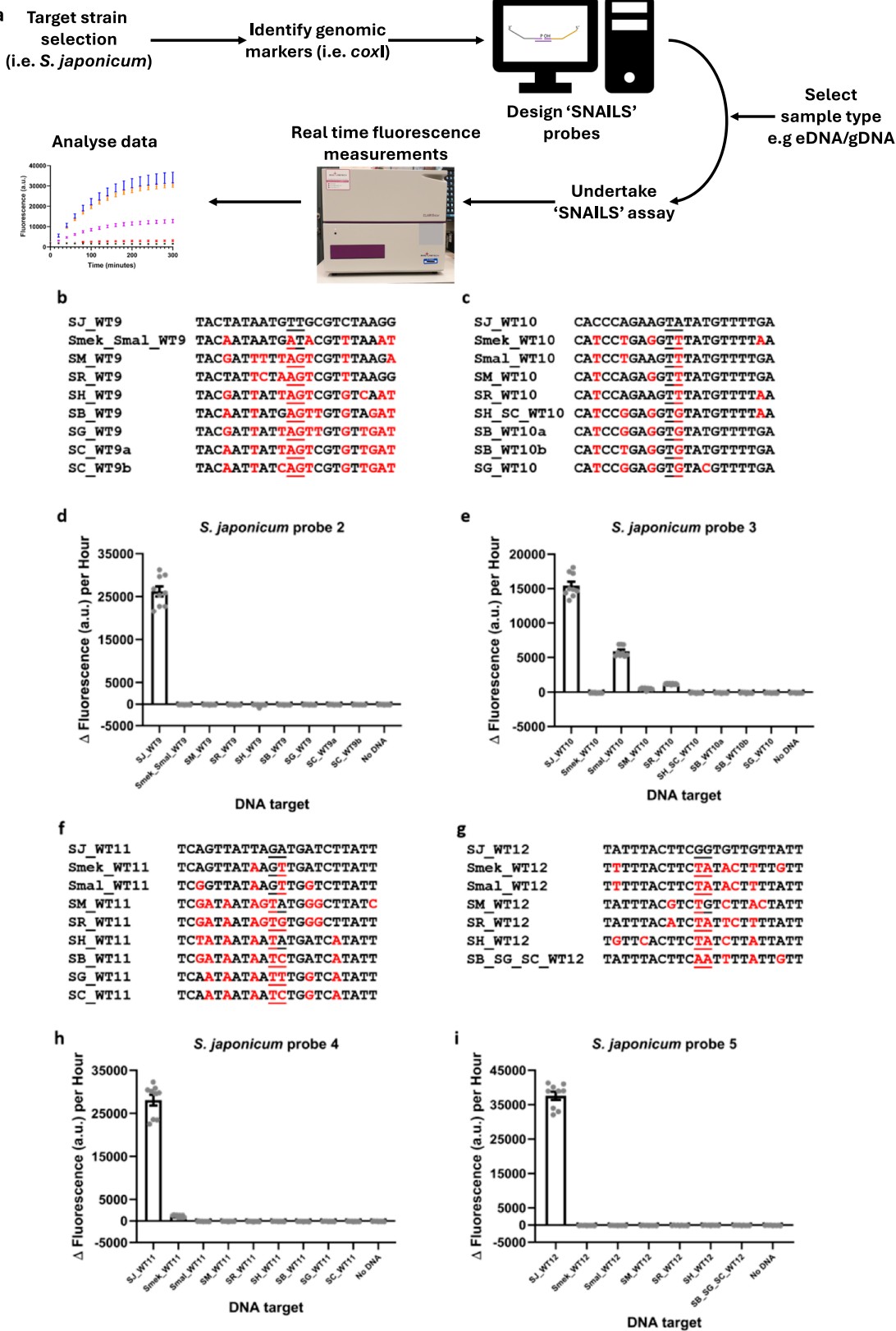

**Communications Biology**| (2025)8:1321

section. When tested against 30 ng of the ssDNA samples, probe 2 was able to detect all sixteen of the *S. japonicum* cercaria-derived samples SJ1-SJ16 but did not detect the three *S. mansoni* samples SM1-SM3 (Fig. 3; Supplementary Fig. 7), thereby showing that the 'SNAILS' biosensor protocol can be used to detect specifically for *S. japonicum*.

## Discussion

The NTD schistosomiasis is endemic to several regions within Asia, South America, Africa and Southern Europe[1,6]. Schistosomiasis japonica, caused by *S. japonicum*, is found in regions of PRC, the Philippines and Indonesia though it was successfully eliminated from Japan in the mid 1990's[1]. The

**Fig. 1 | Initial screening of *S. japonicum*-specific 'SNAILS' biosensor probes.**
**a** Workflow of the 'SNAILS' biosensor protocol. **b, c, f, g** DNA targets tested against *S. japonicum* probes 2 (**b**), 3 (**c**), 4 (**f**) and 5 (**g**). *Schistosoma* targets are indicated as follows: SJ_WT9, SJ_WT10, SJ_WT11 and SJ_WT12 (*S. japonicum*), Smek_S-mal_WT9 (*S. mekongi* and *S. malayensis*), Smek_WT10, Smek_WT11 and Smek_WT12 (*S. mekongi*), Smal_WT10, Smal_WT11 and Smal_WT12 (*S. malayensis*), SM_WT9, SM_WT10, SM_WT11 and SM_WT12 (*S. mansoni*), SR_WT9, SR_WT10, SR_WT11 and SR_WT12 (*S. rodhaini*), SH_WT9, SR_WT11 and SH_WT12 (*S. haematobium*), SH_SC_WT10 (*S. haematobium* and *S. curassoni*), SB_WT9 and SB_WT11 (*S. bovis*), SB_WT10a (*S. bovis* MH647124),

SB_WT10b (*S. bovis* FJ897160), SB_SG_SC_WT12 (*S. bovis*, *S. guineensis* and *S. curassoni*), SG_WT9, SG_WT10 and SG_WT11 (*S. guineensis*), SC_WT9a (*S. curassoni* AY157210), SC_WT9b (*S. curassoni* AJ519516) and SC_WT11 (*S. curassoni*). **d, e, h, i** *S. japonicum* probe sets 2, 3, 4 and 5 respectively, were tested for their ability to recognise the corresponding 22-base *Schistosoma* target regions as indicated. No DNA corresponds to the probe negative control i.e. no target DNA. Probe sets and target concentrations were tested at 50 nM. *n* = 9 (3 replicates per reaction, each reaction split into triplicate runs). Error bars denote standard error of the mean. Probe sequences are available in Supplementary Table 1 and Δ fluorescence (a.u.) per hour values are listed in Supplementary Table 4.

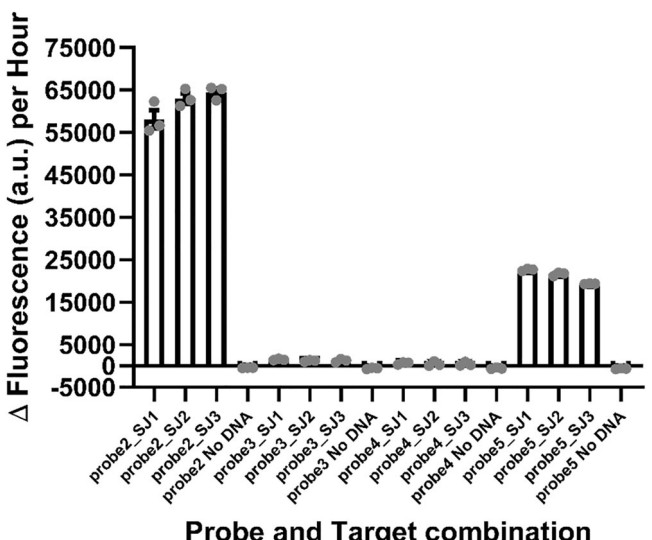

**Fig. 2 | *S. japonicum*-specific probes validation against ssDNA derived from PCR amplification of plasmid DNA.** The ability of *S. japonicum*-specific probes 2, 3, 4 and 5 to bind to ssDNA derived from plasmid DNA. The 446-base *S. japonicum*-specific *cox*1 target region was amplified from plasmid pAJW333 and treated as described in the methods section to produce purified ssDNA. Three PCR reactions were tested against the four probe pairs. 30 ng of target ssDNA was incubated with 50 nM of each probe half. Reactions are identified as follows: probe2_SJ1-3 (probe 2 against ssDNA samples 1-3), probe2 No DNA (probe 2 negative control), probe3_SJ1-3 (probe 3 against ssDNA samples 1-3), probe3 No DNA (probe 3 negative control), probe4_SJ1-3 (probe 4 against ssDNA samples 1-3), probe4 No DNA (probe 4 negative control), and probe5_SJ1-3 (probe 5 against ssDNA samples 1-3), probe5 No DNA (probe negative control). *n* = 3 (each reaction split into triplicate runs). Error bars denote standard error of the mean. Probe sequences are available in Supplementary Table 1 and Δ fluorescence (a.u.) per hour values are listed in Supplementary Table 5.

elimination of *S. japonicum* in Japan gives hope that it may be possible to reduce its prevalence in other countries or regions as well. Indeed, PRC has made great progress since implementing the National Schistosomiasis Control Program in the 1950's, using snail control, MDA programmes and fully integrated control regimes, including changes in environmental and agricultural practices[10,11,18]. Breaking the infective cycle of *S. japonicum* is complicated by the fact that it is a zoonotic species, with 40 or more domestic and wild mammals also able to act as definitive hosts for this parasitic species[7,8]. Thus, eliminating this disease will have great implications for not only human health, but also for broader economic and food security situations. Therefore, the continued development and refinement of species-specific *Schistosoma* diagnostic technologies are critical.

Historically, MDA has been the main control intervention and consequently the majority of diagnostics have been developed for the detection of schistosomes in human and veterinary relevant animal populations. Currently, the Kato-Katz microscopic method[19] of looking for *S. japonicum* eggs in the smears of faecal samples is the primary diagnostic used in PRC

and the Philippines. Although, as in other geographical regions, there are notable limitations to using this technique, with epidemiological studies in both PRC and the Philippines reporting underestimation of prevalence[20,21]. Enzyme-linked immunosorbent assays (ELISA) using either crude or recombinant antigens have also been successfully used as diagnostics for *S. japonicum* infection in humans and water buffalo[22–25]. However, to address the elimination of schistosomiasis it is apparent that a switch to include environmental monitoring is required[26]. As such, as well as monitoring for infections in local human populations currently infected and at risk of exposure, monitoring of other mammalian reservoirs of infection such as water buffalo and snail (*Oncomelania*) surveys should be undertaken[27]. This monitoring could be anatomical/physical, i.e., looking for the presence and positively identifying eggs in the faeces and cercariae released from *Oncomelania* snails or, if the snails show no signs of infection i.e. shedding cercariae, they can be analysed for prepatency (presence of sporocysts and/or cercariae), where they are crushed, usually between microscope slides, and analysed microscopically[28,29].

Indeed, this monitoring could also be molecular based, with nucleic acids such as DNA being the most promising targets. For example, samples from crushed collected snails could be analysed using PCR-based methods[30–32] or loop-mediated isothermal amplification (LAMP) assays[33,34]. Furthermore, DNA can be extracted from parasite samples such as isolated adult worms, miracidia, cercariae and eggs, or as circulating cell-free DNA in blood, urine and faeces[35]. Also, recent focus has been on the analysis of parasite-specific environmental DNA (eDNA) from water or soil samples[36,37], with collection methods for these samples somewhat straightforward compared to extraction methods for releasing DNA from worm or other biological samples. Samples can then be analysed for the presence of schistosomes via PCR[38,39], real-time PCR[40–42], real-time quantitative PCR (qPCR)[35], multi-plex RT-PCR[43], droplet digital PCR (dd-PCR)[21], oligo-chromatographic dip-sticks[41], recombinase polymerase amplification fluorescence assay (RPA)[44,45], LAMP assays[46,47] and DNA sequencing[48]. All of these techniques are useful tools, especially as there has historically been a significant underinvestment in schistosomiasis diagnostics[49]. However, DNA sequencing is expensive and probes for other methods, such as RT-PCR, may not be able to detect and differentiate very small changes in a target DNA sequence such as single nucleotide polymorphisms (SNPs).

Previously, we developed a DNA-based biosensor termed 'SNAILS' for detecting and differentiating *S. mansoni* and *S. haematobium*[13] and in this current study, we have shown that the 'SNAILS' biosensor technology can be successfully applied to the specific detection of *S. japonicum* cercariae samples collected from a range of geographical locations in PRC. Collectively, these two studies now demonstrate that 'SNAILS' biosensors can be used to detect for the three most prevalent *Schistosoma* species that infect humans (*S. mansoni*, *S. haematobium* and *S. japonicum*). However, the current studies shift in focus towards *S. japonicum* necessitated meaningfully different approaches than our previous study, including the collection, sequencing and bioinformatic analyses of new *S. japonicum* samples from appropriate geographical regions (PRC, the Philippines, Japan and Taiwan) which were beyond the scope of our previous Africa-focused schistosomiasis biosensor project. These data also informed the design and validation of several new 'SNAILS' biosensor probes for the species-specific detection of *S. japonicum*. Likewise, 'SNAILS' biosensor workflows also had

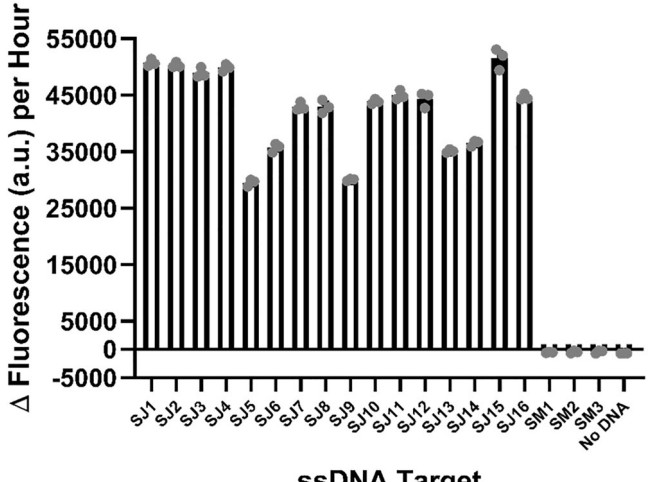

**Fig. 3 | *S. japonicum*-specific probe 2 can detect *S. japonicum* cercariae samples isolated from sites in the People's Republic of China.** The ability of *S. japonicum*-specific probe 2 to recognise and bind to ssDNA derived from *S. japonicum* cercarial gDNA. The 446-base *cox*1 target region was amplified from 16 different *S. japonicum* gDNA samples (SJ1-SJ16) and 3 different laboratory obtained *S. mansoni* gDNA samples (SM1-SM3) and treated as described in the methods section to produce purified ssDNA. 30 ng of target ssDNA was incubated with 50 nM of each probe half (SJ_A2 and SJ_B2). Reactions are identified as follows: SJ1-SJ16 correspond to probe 2 against *S. japonicum* ssDNA samples 1-16, SM1-SM3 corresponds to probe 2 against *S. mansoni* gDNA samples 1-3, No DNA corresponds to probe 2 negative control (no ssDNA target). *n* = 3 (each reaction split into triplicate runs). Error bars denote standard error of the mean. Probe sequences are available in Supplementary Table 1 and Δ fluorescence (a.u.) per hour values are listed in Supplementary Table 7.

to be adapted beyond the use of adult worm gDNA samples to now also enable the utilisation of cercariae samples taken from real-world field sites in PRC. The testing of these new samples also further demonstrates the potential utility of the 'SNAILS' biosensor for use in future schistosomiasis environmental DNA screening studies. Whilst we envision that 'SNAILS' biosensor workflows could potentially be adapted to also incorporate testing of human faeces or urine samples, these types of samples were not available to, nor were they directly relevant to, the present study. This is primarily due to our studies core focus on environmental sampling for schistosomiasis research applications.

Importantly, the 'SNAILS' biosensor technology can detect for very small changes in the 22-base long target DNA sequence, including SNPs[13]. Therefore, future studies could apply this biosensor technology to aid in the detection of genomic sequence variations including those implicated in the resistance to drugs such as oxamniquine (OXA)[50,51] and praziquantel (PZQ)[52], two drugs used extensively for preventing and treating schistosomiasis. Both OXA and PZQ resistance has been associated with loss-of-function mutations, including SNPs, within either the sulphotransferase SmSULT-OR (Smp_089320) for OXA resistance[51], or within the parasite ion channel from the transient receptor potential melastatin family TRPM$_{PZQ}$ for PZQ[52]. Thus, the 'SNAILS' biosensor technology could aid in epidemiological studies for basic research, monitoring effectiveness of WASH interventions and MDAs, as well as potentially detecting for the presence and/or emergence of drug-resistant populations of schistosomes.

PCR amplification is a crucial step for the 'SNAILS' biosensor assay. It both enables the amplification of target DNA from a range of samples, which could include urine, faeces and blood from human patients and animals such as Water Buffalo, or environmental samples such as freshwater, whilst, through the use of a phosphorylated 3' primer, enables the production of the ssDNA target required for the assay via a lambda exonuclease treatment step. Beneficially, the PCR step could allow for positive detection of samples, both biological or environmental, with low parasite levels, which may currently go undetected by current methods that do not incorporate an amplification step.

Indeed, if the PCR amplification step is undertaken using a common multi-species primer set, detection with various probes could be applied to a range of samples, such as clinically relevant samples or environmental samples, for simultaneous multiple detection, thereby creating a multi-species *Schistosoma* panel assay, which the World Health Organisations' (WHO) diagnostic technical advisory group has identified as a need in its latest target product profile (TPP) for both monitoring and evaluation of schistosomiasis control and for interruption and surveillance of schistosomiasis transmission[53]. Furthermore, we believe the 'SNAILS' biosensor could be applied to wide range of other targets (Fig. 4) and could ultimately be used as the base technology platform for the detection of other pathogens relevant to human and veterinary health, either as a single target biosensor or as a panel assay for multiple target/pathogen detection.

## Methods

### Identification of *S. japonicum*-specific *cox*1 targets and probe designs

Previously[13], we used MUSCLE[16] to align the following nucleotide sequences to identify 22-base long species-specific *cox*1 targets for our DNA-based 'SNAILS' biosensor assay: *S. japonicum* EU325878 (GenBank), *S. malayensis* EF635956[54], *S. mekongi* EF635955[54], *S. mansoni* AJ519524[55] and MG562513 (GenBank), *S. rodhaini* AY157202[56], *S. haematobium* GU257336 and GU257338[57], *S. bovis* MH647124[58] and FJ897160[59], *S. guineensis* AJ519517, AJ519522 and AJ519223[55] and *S. curassoni* AJ519516[55] and AY157210[56]. This enabled the identification of *S. japonicum*-specific *cox*1 targets. A second MUSCLE[16] alignment, using default parameters, of EU325878 and other unpublished and published (KR855674[48] and EF635954[54]) *S. japonicum cox*1 nucleotide sequences obtained from GenBank was undertaken, to further validate the *S. japonicum*-specific *cox*1 targets identified (Supplementary Fig. 1). Resultant schistosome targets identified and probe designs are listed in Supplementary Table 1. Oligonucleotide primers used as synthetic biosensor targets, for target amplification from genomic or plasmid DNA and for sequencing, as well as those that comprise the biosensor probes were ordered from Integrated DNA Technologies (IDT, USA). Supplementary Table 2 lists the *S. japonicum* nucleotide sequences obtained from GenBank for this study.

### *S. japonicum* and *S. mansoni* cercarial samples
*S. japonicum* cercarial samples previously collected from a range of locations in China and *S. mansoni* cercarial samples isolated from a laboratory-maintained source (Natural History Museum, UK), were obtained from the Schistosomiasis collection at the Natural History Museum, UK (SCAN)[17]. The sites the *S. japonicum* samples were collected from are listed in Supplementary Table 6. Cercariae samples were provided on discs obtained using a Harris punch.

### Purification and elution of *Schistosoma* gDNA
*Schistosoma* cercarial gDNA was obtained using the previously published combined CGP extraction protocol[60,61]. Briefly, to each of the individual Whatman FTA punches in 200 μL PCR tubes, 30 μL of lysis buffer [30 mM Tris-HCl pH 8.0 (Sigma Aldrich), 1.25 μg/mL of Protease reagent (Qiagen; catalogue #19155), 0.5% Tween®-20 (Fisher Scientific; catalogue #BP337-100), and 0.5% IGEPAL® CA-630 (Sigma Aldrich; catalogue #I3021-100)] was added, making sure the disc is fully submerged in the lysis buffer. Samples were incubated in a ProFlex PCR system (Applied Biosystems, Thermo Fisher Scientific, USA) using the following protocol: 1 cycle of 50 °C for 1 h, followed by 1 cycle of 75 °C for 30 min (to inactivate the protease). Extracted DNA solution was then transferred to a fresh tube, with long term storage at −20 °C.

### Amplification of *Schistosoma* gDNA
Polymerase chain reactions (PCR) were used to amplify the 446-base *cox*1 target region. 1 μL of gDNA (for all cercarial samples) or 1 μL of plasmid DNA (pAJW333; 10 ng DNA total) were used as templates. Cercariae were amplified using either primer pair 5-SJ-cox1/3-SJ-cox1 (AJW1061/

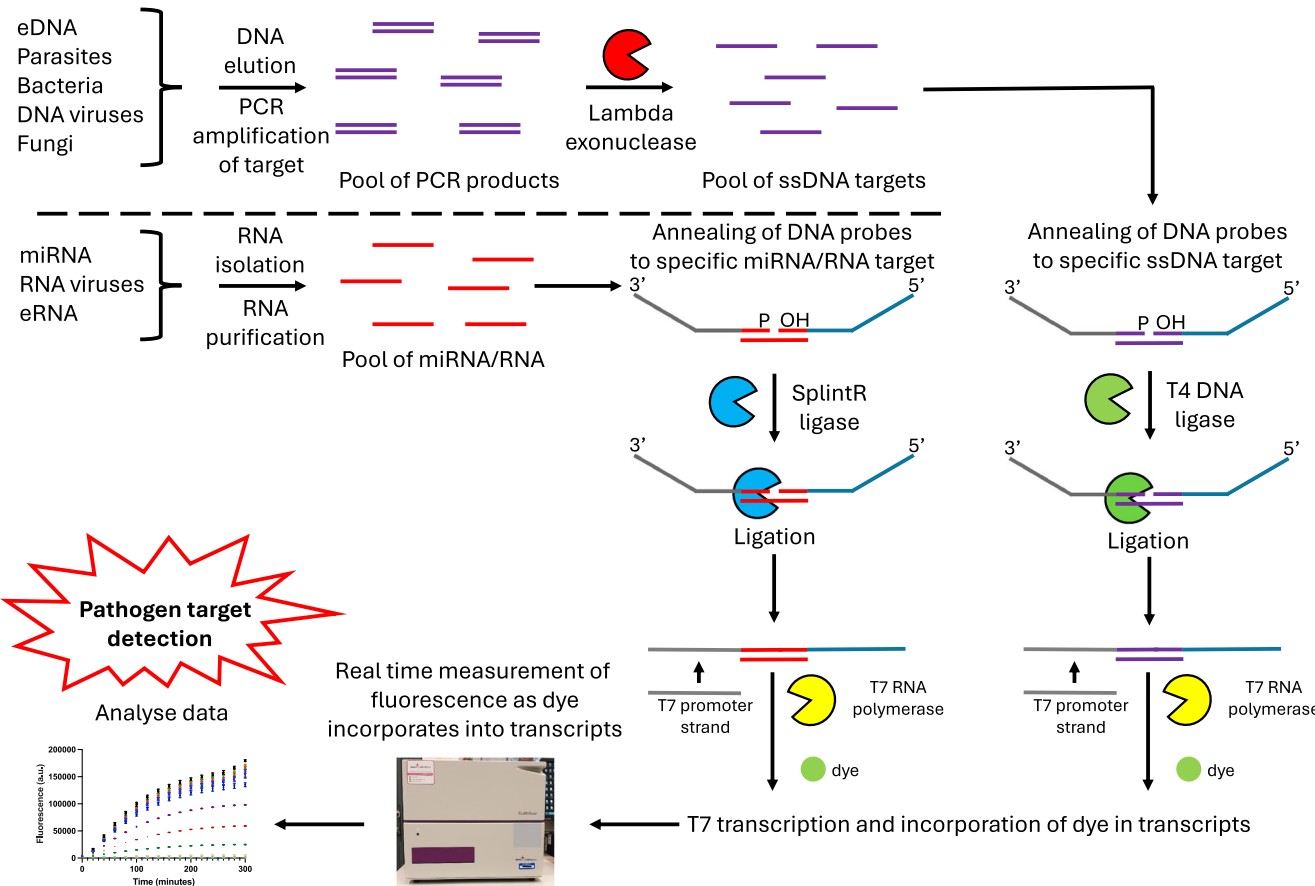

**Fig. 4 | Application of the 'SNAILS' biosensor technology to the detection of other targets.** The 'SNAILS' biosensor platform can be adapted to detect for a range of different pathogens or targets such as drug resistance genes. Not only can DNA targets be detected, but by replacing T4 DNA ligase with SplintR ligase, the DNA probes can anneal to and ligate together when bound to miRNA/RNA targets[14].

AJW1062; *S. japonicum* – specific) or primer pair 5-SM-cox1/3-SM-cox1 (AJW1064/AJW1065; *S. mansoni* – specific) for subsequent sequencing of the *cox*1 446-base target region. For the production of DNA phosphorylated on the 3' strand for downstream 'SNAILS' assay analysis, *S. japonicum* cercariae and control plasmid samples were amplified using primer pair -SJ-cox1/p-3-SJ-cox1 (AJW1061/AJW1063), whilst *S. mansoni* cercariae samples were amplified using primer pair 5-SM-cox1/p-3-SM-cox1 (AJW1064/AJW1066). DNA was amplified using Q5 high-fidelity DNA polymerase (catalogue #M0491S; New England Biolabs, USA) following the manufacturer's instructions for 50 µL reactions. Essentially, 1 µL of relevant DNA sample was amplified using 2.5 µL (10 µM working stock) each of the relevant 5' and 3' primer (see above) in a ProFlex PCR system (Applied Biosystems, Thermo Fisher Scientific, USA) with the following protocol: 1 cycle of 98 °C for 30 seconds, 40 cycles of 98 °C for 10 s, 57 °C for 30 s and 72 °C for 1 min, followed by 1 cycle of 72 °C for 2 min.

DNA samples to be sequenced were extracted from agarose gels and purified using the Monarch DNA gel extraction kit (catalogue #T1020; New England Biolabs, USA) following the manufacturer's instructions, with the DNA eluted in 20 µL of nuclease-free water. Purified DNA samples were sequenced by Eurofins Genomics GmbH (Ebersberg, Germany). The resulting sequences are detailed in Supplementary Table 6 and those corresponding to the *S. japonicum* samples have been deposited in the European Nucleotide Archive under the accessions OZ203289-OZ203304: (http://www.ebi.ac.uk/ena/data/view/<ACCESSION NUMBERS>).

DNA samples for downstream analysis in the 'SNAILS' assay were validated using agarose gel electrophoresis. 5 µL of PCR product was separated using a 0.8% agarose gel with SYBR safe DNA stain (catalogue #S33102; Invitrogen, Thermo Fisher Scientific, MA, USA). Bands in the agarose gel were visualised with the Gel Doc XR+ system with Image Lab software v.6.1 (model #Universal Hood II; Bio-Rad Laboratories Inc., USA).

### Cloning of *S. japonicum* 446-base *cox*1 fragment into pCR-Blunt-II-TOPO DNA

A 446-base pair *cox*1 fragment, corresponding to the 446-base target region indicated in Supplementary Fig. 1, was ordered from Integrated DNA Technologies (IDT, USA) and cloned into the pCR-Blunt-II-TOPO plasmid following the manufacturer's instructions (catalogue #450245; Invitrogen, Thermo Fisher Scientific, MA, USA). The reaction was then transformed into chemically competent *Escherichia coli* NEB10-beta cells (catalogue #C3019H; New England Biolabs, USA). For plasmid recovery, individual colonies were grown in Luria-Bertani (LB) medium supplemented with 35 µg/mL Kanamycin and cultured at 37 °C with shaking (200 rpm). Plasmid DNA was isolated and purified using the QIAprep spin miniprep kit as per the manufacturer's instructions (catalogue #27106; Qiagen, UK). The DNA sequence of the insert was verified using the sequencing services provided by Full Circle Labs Ltd (London, UK). The resulting plasmid (pAJW333) is detailed in Supplementary Table 3.

### Generation of ssDNA

Lambda exonuclease (catalogue #EN0562; Thermo Fisher Scientific, MA, USA) was used to generated the 446-base long ssDNA as follows: to the 45 µL PCR reactions, 1 µL Lambda exonuclease (10 U/µL), 6 µL 10X reaction buffer [670 mM glycine-KOH (pH 9.4), 25 mM MgCl₂, 0.1% (v/v) Triton X-100] and 8 µL nuclease-free H₂O were added to produce a 60 µL reaction volume. The reactions were gently mixed via pipetting and incubated in a ProFlex PCR system (Applied Biosystems, Thermo Fisher

Scientific, USA) using the following protocol: 1 cycle at 37 °C for 30 min, 1 cycle at 80 °C for 10 min (to inactivate the lambda exonuclease) and then a final cycle at 10 °C. The ssDNA produced was purified using the Monarch PCR and DNA cleanup kit (5 µg) (catalogue #T1030; New England Biolabs, USA) as per the manufacturer's instructions for ssDNA, with the samples being eluted in 20 µL of nuclease-free H$_2$O. The concentration of these samples was measured, according to the manufacturer's instructions, using a Qubit 3 (Fluorometer Thermo Fisher Scientific, MA, USA) and a Qubit ssDNA assay kit (catalogue #Q10212; Thermo Fisher Scientific, MA, USA).

### SNAILS assay
The 'SNAILS' biosensor assay was undertaken as previously published[13]. For the probe and target annealing reactions a final concentration of 50 µM of each probe half was added to either 50 µM of the 22-base synthetic target or 30 ng of the ssDNA generated from PCR reactions. Concentrations of probes and targets are based on the 10 µL volume of the ligation reaction step of the 'SNAILS' assay.

Annealing of probes and target ssDNA: Annealing reactions (7 µL) comprised 1 µL half probe A, 1 µL half probe B, 1 µL target ssDNA (or 1 µL nuclease-free H$_2$O for negative controls) and 4 µL nuclease-free duplex buffer (30 mM HEPES, pH 7,5; 100 mM CH$_3$CO$_2$K; catalogue #11-01-03-01, IDT, USA). Reactions were mixed gently via pipetting, incubated at 95 °C for 3 min in a ProFlex PCR system (Applied Biosystems, Thermo Fisher Scientific, USA) and then incubated on ice for 5 min.

Ligation of probes: To the annealing reactions (7 µL), 1 µL nuclease-free H$_2$O, 1 µL T4 DNA ligase (400 U; catalogue # M0202L; New England Biolabs) and 1 µL 10X T4 DNA ligase buffer [final 1X concentration in reaction: 50 mM Tris-HCL, 10 mM MgCl$_2$, 1 mM ATP, 10 mM DTT, pH 7.5] were added, mixed gently via pipetting and incubated at 25 °C for 20 min, 65 °C for 10 min (to inactivate T4 DNA ligase) and then 25 °C for 5 min in a ProFlex PCR system (Applied Biosystems, Thermo Fisher Scientific, USA).

T7 transcription and fluorescence detection: Transcription reactions (60 µL final volume) using the TranscriptAid T7 high yield transcription kit (catalogue #K0441; Thermo Scientific, Thermo Fisher Scientific, USA), comprised the following: 1.5 µL RNase Inhibitor (60 U; catalogue #M0314L; New England Biolabs), 24 µL NTP mix (final concentration of 10 mM each of ATP, CTP, GTP and UTP), 12 µL 5X TranscriptAid reaction buffer, 6 µL dithiothreitol (DTT; 0.1 M solution; catalogue #707265ML; Thermo Scientific, Thermo Fisher Scientific, USA), 6 µL TranscriptAid enzyme mix, 3 µL 2$^{nd}$ T7 promoter sequence oligonucleotide (1 µM stock solution), 6.3 µL of the ligation product and 1.2 µL DFHBI-1T (10 µM final concentration; catalogue #5610, Tocris Bioscience). Reactions were mixed gently via pipetting and three 15 µL aliquots of each reaction were loaded into 384-well plates (catalogue #781096, Greiner Bio-one, Kremsmünster, Austria). Reactions were measured using a CLARIOstar plate reader (BMG, Ortenberg, Germany) with the following settings: Excitation 440-15nM/Emission 510-20 nM, orbital shaking for 5 seconds at 500 rpm before measurement every 5 min. Delta (∆) fluorescence measurements (a.u.) per hour were calculated between 20 and 80 min of the reaction runs.

### Statistics and Reproducibility
Statistical analysis (standard error of the mean (s.e.m)) was carried out using GraphPad Prism 10.4.1 (GraphPad Software Inc., La Jolla, California). Reproducibility: for Fig. 1, $n = 9$ (3 replicates per reaction, each reaction split into triplicate runs); for Figs. 2 and 3, $n = 3$ (each reaction split into triplicate runs).

### Reporting summary
Further information on research design is available in the Nature Portfolio Reporting Summary linked to this article.

### Data availability
Data underlying Figs. 1, 2 and 3 can be found in the Supplementary data excel file. All other data is available in the supplementary information file.

*Schistosoma japonicum* sequences are available in the supplementary information file and the European Nucleotide Archive, under accession codes OZ203289-OZ203304, as indicated in the methods section of the main text. Other data is available from the authors upon request.

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

## Acknowledgements
We would like to thank colleagues in the Section of Structural and Synthetic Biology within the Department of Infectious Disease at Imperial College London for their advice and helpful comments. AJW and PSF received funding from the Imperial College London Engineering and Physical Sciences Research Council (EPSRC) Impact Acceleration Account [EP/R511547/1]. RJRK and PSF received funding through the Biotechnology and Biological Sciences Research Council (BBSRC) grant [BB/W012987/1]. AJW, RJRK and PSF received funding from the EPSRC via SynbiCITE proof of concept funding [EP/Z533142/1], and the EPSRC/UKRI/Future Biomanufacturing Research Hub (FBRH) at The University of Manchester [EP/S01778X/1].

## Author contributions
All authors conceptualised the overall study. A.J.W. performed the experiments. Q.Z. and A.M.E. provided the *Schistosoma* cercariae samples. A.J.W., F.A., R.J.R.K., and A.M.E. analysed data. P.S.F., and A.M.E. supervised the project. A.J.W. wrote the paper, all authors edited the paper.

## Competing interests
The authors declare no competing interests.
