## [Transparent Peer Review file · Communications Biology]

Species-specific detection of *Schistosoma japonicum* using the 'SNAILS' DNA-based biosensor

Corresponding Author: Professor Paul Freemont

This manuscript has been previously submitted at another journal. This document only contains information relating to versions considered at Communications Biology.

Version 0:

Reviewer comments:

Reviewer #1

(Remarks to the Author)

This is a well written and organized manuscript. It takes advantage of a well-established technique that was developed and validated by the authors termed SNAILS to detect *S. mansoni* and *S. haematobium* which co-occur in sub-Saharan Africa. They now use the technique to identify *S. japonicum* (Sj) that is a zoonotic form of schistosomiasis that can infect mammals including humans. They provide sufficient information including a figure, for investigators to perform the technique. They developed 4 probes against the Sj cox1 target region (sequences included). They demonstrated that probe 2 could identify the target region in 16 different Sj cercarial gDNA but not *S. mansoni* gDNA.

There are a number of geographical strains of Sj eg. Philippines, Chinese, Formosan. It would have been relevant to test probe 2 against these strains that are transmitted through different subspecies of *Oncomelania hupensis*. Likewise, the authors claim the SNAIL technique would be useful in diagnosing drug resistance. Both oxamniquel and praziquantel resistant strains which are double recessives are available for testing.

Overall, this is a significant advance for the diagnosis of human species of *Schistosoma*. In the case of Sj it is relevant for diagnosing animals such as water buffalo for Sj. As the authors argue, it may have uses in the diagnosis of other infections.

Reviewer #2

(Remarks to the Author)

In this manuscript, authors developed and refined the 'SNAILS' DNA-based biosensor technology for specific detection of *Schistosoma japonicum*. However, I don't think the final purpose of the manuscript is clear. In this study, authors didn't verify the specificity of the method for detecting *S. japonicum*, and the sensitivity of the method was also unclear. Which kinds of the samples from host could be used for detecting? Meanwhile, authors also didn't use some clinical samples to verify the effect of the method. Moreover, I also have some other questions. What's the cost of the detection for each sample? What are the application scenarios and the detection subjects of the 'SNAILS' method in the future? Why did authors not find some special targets for molecular detection from *S. japonicum* and set up some saving time methods, such as qPCR. I think it will finally achieve this aim.

Reviewer #3

(Remarks to the Author)

In this paper, the authors developed and refined the DNA-based biosensor technology "SNAILS" for the detection of schistosomiasis and applied it for specific detection of *S. japonicum*. The results seem reasonable. However, there were some issues that needed to be addressed before final acceptance for publication. I suggest that the following issues be resolved before further analysis

1-The abstract does not provide clear information about the results obtained in the optimized project. The detection limits for genomic DNA isolated from *Schistosoma mansoni* cercariae should be added

2-Units of concentration should be standardized throughout the manuscript. In some sections, concentration is reported as µg/mL, while in others it appears simply as µg, for example.

3-Please provide more detailed information regarding the transcription and fluorescence assay procedures

4-Consider providing more detailed information about the statistical analyses performed. Although the software (GraphPad Prism) and the use of (s.e.m) are mentioned, it is not clear what specific statistical tests were applied to the data.

5-To strengthen the applicability and robustness of the developed SNAILS DNA-based biosensor, I recommend including storage stability tests. Evaluating how the sensor performs over time under various storage conditions is crucial for assessing its shelf life.

Version 1:

Reviewer comments:

Reviewer #1

(Remarks to the Author)

The authors have addressed each comment in great detail. In fact, they are to be commended for their answers. I have no further concerns and recommend acceptance.

Reviewer #2

(Remarks to the Author)

Thanks authors for response to my comments. However, I still think that authors need to add some clinical samples to verify the method, and it's necessary. Authors can use some other certain organisms' genomic DNA to verify the specificity of the method. In clinical samples, these genomic DNA of some uncertain organisms are complex. Thus, authors could not to make sure that the specificity of the method are excellent. Authors also mentioned that 'this method can detect and differentiate small changes in the short target sequence'. This also means if some small changes in the target sequence, the sample might be missing out. How did authors to make sure the conservation of these targets in *S. japonicum* from all epidemic areas? Such as the Philippines strain or some strains from other southeast Asian countries. In authors' previous study, they tested specificity of the method for *S. mansoni*. In this study, authors used *S. japonicum* as the research object, it also needed to retest the specificity of these targets of *S. japonicum* distinguishing from some other organisms. Meanwhile, authors also provided the data that the detection limit of genomic DNA of *S. japonicum*. Did authors only change the research object (*S. japonicum*) in this study compared to their previous study (*S. mansoni*)? Why did authors not use the method in the previous study to detect clinical samples? Such as water, feces or urine.

Reviewer #3

(Remarks to the Author)

The manuscript has reached a publishable standard following thorough revision and refinement by the authors. I therefore recommend it for publication and commend the authors for their efforts.

Editor and Reviewer comments:

Editor/Referee comment	Author response
Editor comment 1: “For all graphs depicting a single point value (e.g., mean) with error bars, you must add individual data points or convert the graph to a boxplot or dot-plot to show data distribution.”	Thank you, in response we have amended the relevant graphs to show the individual data points. The revised figures are attached below this table.
Editor comment 2: “It’s mandatory to provide access to the numerical source data for graphs and charts either through a repository or by providing the data in a Supplementary Data file (in excel format).”	The numerical source data for the fluorescence time courses indicated in Supplementary Figures 2, 3, 4, 5, 6 and 7 are now provided in an excel spreadsheet labelled “Supplementary Data file”. The delta fluorescence values for the bar graphs in the main text (Figures 1d, e, h, i; Figure 2; Figure 3) are calculated from these time courses and these values, along with the mean and the standard error of the mean (SEM) are fully provided in Supplementary Tables 4, 5 and 7.
Editor comment 3: “All blots/gels must be accompanied by size markers in every figure panel. Uncropped and unedited blot/gel images must be included as Supplementary Figure(s) in the Supplementary Information pdf.”	Not applicable. No blots/gels are provided as part of this manuscript.
Editor comment 4: “Please ensure that you have complied with the data deposition policies at the Nature Portfolio, please see here.”	We have, the Schistosoma japonicum cox1 sequences obtained for this study have been submitted to and deposited in the European Nucleotide Archive (ENA) under the accessions OZ203289-OZ203304.
Editor comments 5 and 6: “Please ensure that you have complied with our policies on research involving animals and humans, see here.” “Please follow the ARRIVE guidelines for reporting animal experiments. Please fully complete an ARRIVE checklist including both the essential and recommended set of items (adding information to the manuscript where needed) and upload this with your revised manuscript.”	We have. No animals or human samples were utilised in this study.
Reviewer 1 comment 1: “This is a well written and organized manuscript. It takes advantage of a well-established technique that was developed and validated by the authors termed SNAILS to detect S. mansoni and S. haematobium which co-occur in sub-Saharan Africa. They now use the technique to identify S. japonicum (Sj) that is a zoonotic form of schistosomiasis that can infect mammals including humans.”	We thank the reviewer for their overall positive and supportive appraisal of our manuscript.
Reviewer 1 comment 2: “They developed 4 probes against the Sj cox1 target region (sequences included). They demonstrated that probe 2 could identify the target region in 16 different Sj cercarial gDNA but not S. mansoni gDNA. There are a number of geographical strains of Sj eg. Philippines, Chinese, Formosan. It would have been relevant to test probe 2 against these strains that are transmitted through different subspecies of Oncomelania hupensis .”	We thank the reviewer for their positive comment: “They demonstrate that probe 2 could identify the target region in 16 different Sj cercarial gDNA but not S. mansoni gDNA.” We further thank the reviewer for highlighting this important point about different geographical strains of S. japonicum i.e. those found in China versus those isolated from the Philippines etc. For this study, when we further interrogated the potential probe targets we previously identified for S. japonicum , the available cox1 FASTA sequences we used were isolated and obtained from different geographical locations as indicated in Supplementary Table 2 (both China and the Philippines).

	However, in response to the reviewer 1's comment, we further interrogated GenBank and found further cox1 FASTA sequences for S. japonicum isolated in Japan and Taiwan (see Supplementary Table 2). We have also repeated the sequence alignment using MUSCLE and the updated set of cox1 FASTA sequences and have updated Supplementary Figure 1 accordingly. The four probe targets we identified are present in all of the cox1 FASTA sequences analysed by us in this study. This, therefore, indicates that the probes in this study are designed to negate this geographical variation in strains and are designed to detect for S. japonicum isolated from different geographical regions (China, the Philippines, Japan and Taiwan). To further clarify this, we have edited the manuscript as follows: Lines 110-113: "These included sequences obtained from a range of locations in PRC, the Philippines, Japan and Taiwan, thereby enabling the identification of S. japonicum targets covering a range of geographical strains/isolates of this species. Four of these targets were selected for further molecular analysis." The method section was also changed as follows: Lines 333-337: "A second MUSCLE¹⁶ alignment, using default parameters, of EU325878 and other unpublished and published (KR855674⁴⁸ and EF635954⁵²) S. japonicum cox1 nucleotide sequences obtained from GenBank was undertaken, to further validate the S. japonicum-specific cox1 targets identified (Supplementary Figure 1)."
Reviewer 1 comment 3: "Likewise, the authors claim the SNAIL technique would be useful in diagnosing drug resistance. Both oxamniquine and praziquantel resistant strains which are double recessives are available for testing."	We thank the reviewer for their comment on this discussion point. We do believe that the 'SNAILS' DNA-based biosensor technology has the inherent ability to be applied to the detection of many different targets, including other pathogens as well as for specific gene targets such as diagnosing drug resistance. Although we do mention in the discussion the possible application of the 'SNAILS' technology in the detection of oxamniquine (OXA) resistance, testing the 'SNAILS' technology against OXA or praziquantel (PZQ) resistant Schistosoma samples would be a separate study altogether, and out of the scope of this current study. Indeed, OXA resistance has been associated with loss-of-function mutations, including single nucleotide polymorphisms (SNPs), within the sulphotransferase SmSULT-OR (Smp_089320) whilst PZQ resistance has been associated with SNPs within the parasite ion channel from the transient receptor potential melastatin family TRPM_{PZQ}. As the design of the 'SNAILS' biosensor technology allows for the detection of very small changes in the 22-base long target sequence, including SNPs, then in the future the 'SNAILS' biosensor technology could be applied to the detection of resistance associated SNPs either individually, or as a panel assay with multiple probes. We have edited the manuscript to further discuss and clarify this: Lines 279-294: "However, DNA sequencing is expensive and probes for other methods, such as RT-PCR, may not be able to detect and differentiate very small changes in a target DNA sequence such as single nucleotide polymorphisms (SNPs). "Previously, we developed a DNA-based biosensor termed 'SNAILS' for detecting and differentiating S. mansoni and S. haematobium¹³ and in this current study, we have shown that the 'SNAILS' biosensor technology can be successfully applied to the specific detection of S. japonicum cercariae samples collected from a range of geographical locations in PRC.

	Importantly, the ‘SNAILS’ biosensor technology can detect for very small changes in the 22-base long target DNA sequence, including SNPs¹³. Therefore, future studies could apply this biosensor technology to aid in the detection of genomic sequence variations including those implicated in the resistance to drugs such as oxamniquine (OXA)^{50,51} and praziquantel (PZQ)⁵², two drugs used extensively for preventing and treating schistosomiasis. Both OXA and PZQ resistance has been associated with loss-of-function mutations, including SNPs, within either the sulphotransferase SmSULT-OR (Smp_089320) for OXA resistance⁵¹, or within the parasite ion channel from the transient receptor potential melastatin family TRPM_{PZQ} for PZQ⁵².” Also, lines 311-315: Furthermore, we believe the ‘SNAILS’ biosensor could be applied to wide range of other targets (Fig. 4) and could ultimately be used as the base technology platform for the detection of other pathogens relevant to human and veterinary health, either as a single target biosensor or as a panel assay for multiple target/pathogen detection.”
Reviewer 1 comment 4: “Overall, this is a significant advance for the diagnosis of human species of Schistosoma. In the case of Sj it is relevant for diagnosing animals such as water buffalo for Sj. As the authors argue, it may have uses in the diagnosis of other infections.”	We thank the reviewer for their appraisal that our study “is a significant advance for the diagnosis of human species of Schistosoma.” We also agree that “In the case of Sj it is relevant for diagnosing animals such as water buffalo for Sj.” We have further clarified this point as follows: Lines 298-302: “PCR amplification is a crucial step for the ‘SNAILS’ biosensor assay. It both enables the amplification of target DNA from a range of samples, which could include urine, faeces and blood from human patients and animals such as Water Buffalo, or environmental samples such as freshwater, whilst, through the use of a phosphorylated 3’ primer, enables the production of the ssDNA target required for the assay via a lambda exonuclease treatment step.”
Reviewer 2 comment 1: “In this manuscript, authors developed and refined the ‘SNAILS’ DNA-based biosensor technology for specific detection of Schistosoma japonicum. However, I don’t think the final purpose of the manuscript is clear.”	We thank the reviewer their comments. Originally, the ‘SNAILS’ DNA-based biosensor was successfully developed to aid in the environmental screening and detection of Schistosoma mansoni and Schistosoma haematobium as part of the Water Infrastructure for Schistosomiasis-Endemic Regions (WISER) project. As documented, there has historically been a significant underinvestment in schistosomiasis diagnostics, and as such the continued development and refinement of species-specific Schistosoma diagnostic technologies are critical. S. japonicum is a zoonotic species that has more than 40 different domestic and wild mammal species able to act as definitive hosts. Furthermore, the disease it causes, schistosomiasis japonica, has higher morbidity than schistosomiasis caused by other species, due to the higher number of eggs released by S. japonicum. As such, and as for other schistosomes of human and veterinary importance, it is desirable for public health reasons to break the infective cycle. We, therefore, decided to examine whether the ‘SNAILS’ DNA-based biosensor technology could be applied to the detection of this parasite. In this study, we were able to successfully apply the ‘SNAILS’ biosensor technology to specifically detect S. japonicum using synthetic targets, and more importantly real-world biological samples (cercariae) isolated from a range of sites in the People’s Republic of China. We believe that the purpose of the manuscript is clear. In the abstract we state: Lines 20-25: “In this study, we have further developed and refined the ‘SNAILS’ DNA-based biosensor

	technology for the detection of schistosomes and have applied this to the specific detection of S. japonicum. Of the four new ‘SNAILS’ probe pairs developed, one of our optimised designs successfully detected and differentiated between genomic DNA isolated from 16 S. japonicum cercariae isolated from sites in the People’s Republic of China and laboratory-derived Schistosoma mansoni cercariae.” Also, Lines 74-76: “Here, we report on the successful application of the ‘SNAILS’ biosensor technology for the species-specific detection of S. japonicum.” And, Lines 282-285: “Previously, we developed a DNA-based biosensor termed ‘SNAILS’ for detecting and differentiating S. mansoni and S. haematobium¹³ and in this current study, we have shown that the ‘SNAILS’ biosensor technology can be successfully applied to the specific detection of S. japonicum cercariae samples collected from a range of geographical locations in PRC.” Although the focus of the ‘SNAILS’ biosensor, in both the previously published proof-of-concept paper and this current study on S. japonicum detection, has been presented from an environmental monitoring prospect, the fact that the assay method requires a PCR amplification step means that this sensor technology could also fairly quickly be applied to analysing human and veterinary relevant clinical samples such as faeces, urine, blood and isolated adult worms, as well as environmental samples such as snails, cercariae and water samples. In this study, we did test “real-world” biologically relevant samples (cercariae) isolated in the field from locations in the People’s Republic of China. The goal of this study was to show that the ‘SNAILS’ DNA-based biosensor could also be applied to S. japonicum, another species of schistosome relevant to human and veterinary health. We believe that future iterations of this biosensor, such as making it a multi-species platform assay will also bring it in line with the WHO organisations Diagnostic Target Product Profiles (DPP). As such, we have modified the discussion as follows: Lines 305-311: “Indeed, if the PCR amplification step is undertaken using a common multi-species primer set, detection with various probes could be applied to a range of samples, such as clinically relevant samples or environmental samples, for simultaneous multiple detection, thereby creating a multi-species Schistosoma panel assay, which the World Health Organisations’ (WHO) diagnostic technical advisory group has identified as a need in its latest target product profile (TPP) for both monitoring and evaluation of schistosomiasis control and for interruption and surveillance of schistosomiasis transmission⁵³.”
Reviewer 2 comment 2: “In this study, authors didn’t verify the specificity of the method for detecting S. japonicum”	We thank the reviewer for their comment. This current study builds on our previous publication: “Specific Nucleic Acid Ligation for the detection of Schistosomes: SNAILS”, in which we showed extensive sensitivity and specificity data. In that study, we showed that the tested proof-of-concept S. mansoni probe set 1 could even distinguish between the target sequence it was designed to detect, and target sequences with single (SNPs), double and multiple base changes. We then showed that the probe could differentiate between the S. mansoni target it was designed to detect and the corresponding target sequence from a range of related schistosome species.

	In this current study we believe that we have verified the specificity of the S. japonicum-specific probe designs. For the four probe designs, we tested their ability to specifically detect (i.e. anneal to) the S. japonicum target they were designed to detect as well as the corresponding target region from a range of related schistosome species, see Figure 1. Furthermore, Figure 1b, c, f and g indicate the differences between the target sequences tested and the S. japonicum target sequence (differences are indicated by red letters). Figure 1 therefore, shows the specificity profiles for the four probes design to detect S. japonicum.
Reviewer 2 comment 3: “and the sensitivity of the method was also unclear.”	We thank the reviewer for their comment. The current study uses our previously developed DNA-based biosensor and applies it to the detection of S. japonicum. In our previous publication, we rigorously tested the optimum concentration of probes and target and found that 50 nM of each probe half and 50 nM of synthetic 22-base target (for testing sensitivity and specificity) was suffice. Therefore, for this study we used those same concentrations. For clarity, we have modified the manuscript to make this clearer: Lines 123-125: “In our previous proof-of-concept study, we determined that 50 nM of each probe half and 50 nM of each 22-base long synthetic target were appropriate for testing¹³.” Also, in the previous study, when testing the biosensor against purified ssDNA, we identified that 30 ng of ssDNA in the final 10 µL ligation reaction of the assay was appropriate. For clarity we have modified the manuscript as follows: Lines 137-139: “Previously, we determined that 30 ng of the purified ssDNA target in the final 10 µL ligation reaction of the biosensor assay was appropriate for testing against 50 nM of each probe half¹³.”
Reviewer 2 comment 4: “Which kinds of the samples from host could be used for detecting?”	As our assay uses a PCR step to both amplify our target of choice and then enable the production of the single stranded DNA needed in the biosensor test, we believe that any sample containing potential target DNA could be tested. We believe that the test could be applied to both clinical end environmental samples, such as faeces, urine, blood, circulating cell-free DNA, parasite eggs, cercariae, miracidiae, worms and water and soil samples amongst others. We mention this in the manuscript: Lines 298-311: “PCR amplification is a crucial step for the ‘SNAILS’ biosensor assay. It both enables the amplification of target DNA from a range of samples, which could include urine, faeces and blood from human patients and animals such as Water Buffalo, or environmental samples such as freshwater, whilst, through the use of a phosphorylated 3’ primer, enables the production of the ssDNA target required for the assay via a lambda exonuclease treatment step. Beneficially, the PCR step could allow for positive detection of samples, both biological or environmental, with low parasite levels, which may currently go undetected by current methods that do not incorporate an amplification step. Indeed, if the PCR amplification step is undertaken using a common multi-species primer set, detection with various probes could be applied to a range of samples, such as clinically relevant samples or environmental samples, for simultaneous multiple detection, thereby creating a multi-species Schistosoma panel assay, which the World Health Organisations’ (WHO) diagnostic technical advisory group has identified as a need in its latest target product profile (TPP) for both monitoring and evaluation of schistosomiasis control and for interruption and surveillance of schistosomiasis transmission⁵³.”

Reviewer 2 comment 5: “Meanwhile, authors also didn’t use some clinical samples to verify the effect of the method.”	For the scope of this study we tested “real-world” biological samples (cercariae) collected from locations in the People’s Republic of China. We do believe that the technology can be applied to clinical samples such as Faeces and urine, but that would be under the scope of a future study. Lines 302-311: “Beneficially, the PCR step could allow for positive detection of samples, both biological or environmental, with low parasite levels, which may currently go undetected by current methods that do not incorporate an amplification step. Indeed, if the PCR amplification step is undertaken using a common multi-species primer set, detection with various probes could be applied to a range of samples, such as clinically relevant samples or environmental samples, for simultaneous multiple detection, thereby creating a multi-species Schistosoma panel assay, which the World Health Organisations’ (WHO) diagnostic technical advisory group has identified as a need in its latest target product profile (TPP) for both monitoring and evaluation of schistosomiasis control and for interruption and surveillance of schistosomiasis transmission⁵³.”
Reviewer 2 comment 6: “Moreover, I also have some other questions. What’s the cost of the detection for each sample?”	Previously, we have calculated that our working biosensor technology costs ~£2.49 (~\$3.35 USD) per sample. Future modifications etc may reduce cost such as by automating parts of the assay, removing certain enzymatic steps (if feasible) and by reducing the working reaction volumes for each sample. However, we believe that this is out of the scope of this current study.
Reviewer 2 comment 7: “What are the application scenarios and the detection subjects of the ‘SNAILS’ method in the future?”	We thank the reviewer for their question. We believe we have addressed this with these modifications to the manuscript: Lines 298-315: “PCR amplification is a crucial step for the ‘SNAILS’ biosensor assay. It both enables the amplification of target DNA from a range of samples, which could include urine, faeces and blood from human patients and animals such as Water Buffalo, or environmental samples such as freshwater, whilst, through the use of a phosphorylated 3’ primer, enables the production of the ssDNA target required for the assay via a lambda exonuclease treatment step. Beneficially, the PCR step could allow for positive detection of samples, both biological or environmental, with low parasite levels, which may currently go undetected by current methods that do not incorporate an amplification step. Indeed, if the PCR amplification step is undertaken using a common multi-species primer set, detection with various probes could be applied to a range of samples, such as clinically relevant samples or environmental samples, for simultaneous multiple detection, thereby creating a multi-species Schistosoma panel assay, which the World Health Organisations’ (WHO) diagnostic technical advisory group has identified as a need in its latest target product profile (TPP) for both monitoring and evaluation of schistosomiasis control and for interruption and surveillance of schistosomiasis transmission⁵³. Furthermore, we believe the ‘SNAILS’ biosensor could be applied to wide range of other targets (Fig. 4) and could ultimately be used as the base technology platform for the detection of other pathogens relevant to human and veterinary health, either as a single target biosensor or as a panel assay for multiple target/pathogen detection.”
Reviewer 2 comments 8: Why did authors not find some special targets for molecular detection from S. japonicum and set up some saving time methods, such as qPCR. I think it will finally achieve this aim.	We thank the reviewer for their comment. Although qPCR and RT-PCR are excellent methods, they may not be able to detect and differentiate very small changes in target DNA sequences such SNPs. The benefit of our biosensor technology is that it can

	detect and differentiate small changes in the short (22-base) target sequence. We highlight these benefits of our biosensor assay in the manuscript as follows: Lines 286-297: “Importantly, the ‘SNAILS’ biosensor technology can detect for very small changes in the 22-base long target DNA sequence, including SNPs¹³. Therefore, future studies could apply this biosensor technology to aid in the detection of genomic sequence variations including those implicated in the resistance to drugs such as oxamniquine (OXA)^{50,51} and praziquantel (PZQ)⁵², two drugs used extensively for preventing and treating schistosomiasis. Both OXA and PZQ resistance has been associated with loss-of-function mutations, including SNPs, within either the sulphotransferase SmSULT-OR (Smp_089320) for OXA resistance⁵¹, or within the parasite ion channel from the transient receptor potential melastatin family TRPM_{PZQ} for PZQ⁵². Thus, the ‘SNAILS’ biosensor technology could aid in epidemiological studies for basic research, monitoring effectiveness of WASH interventions and MDAs, as well as potentially detecting for the presence and/or emergence of drug-resistant populations of schistosomes. And, Lines 305-315: “Indeed, if the PCR amplification step is undertaken using a common multi-species primer set, detection with various probes could be applied to a range of samples, such as clinically relevant samples or environmental samples, for simultaneous multiple detection, thereby creating a multi-species Schistosoma panel assay, which the World Health Organisations’ (WHO) diagnostic technical advisory group has identified as a need in its latest target product profile (TPP) for both monitoring and evaluation of schistosomiasis control and for interruption and surveillance of schistosomiasis transmission⁵³. Furthermore, we believe the ‘SNAILS’ biosensor could be applied to wide range of other targets (Fig. 4) and could ultimately be used as the base technology platform for the detection of other pathogens relevant to human and veterinary health, either as a single target biosensor or as a panel assay for multiple target/pathogen detection.”
Reviewer 3 comment 1: “In this paper, the authors developed and refined the DNA-based biosensor technology “SNAILS” for the detection of schistosomiasis and applied it for specific detection of S. japonicum. The results seem reasonable. However, there were some issues that needed to be addressed before final acceptance for publication. I suggest that the following issues be resolved before further analysis”	We thank the reviewer for their comments and suggestions, and their appraisal that “The results seem reasonable.”
Reviewer 3 comment 2: “The abstract does not provide clear information about the results obtained in the optimized project.”	We have modified the abstract to increase its clarity about the results obtained as follows: Lines 22-26: “Of the four new ‘SNAILS’ probe pairs developed, one of our optimised designs successfully detected and differentiated between genomic DNA isolated from 16 S. japonicum cercariae isolated from sites in the People’s Republic of China and laboratory-derived Schistosoma mansoni cercariae.”
Reviewer 3 comment 3: “The detection limits for genomic DNA isolated from Schistosoma mansoni cercariae should be added.”	In our previous proof-of-concept manuscript we analysed the detection limits for both the S. mansoni probe set 1 and S. haematobium probe set 1. We show the limits for detection in that paper. From that analysis we decided that 30 ng (total) of ssDNA in the 10 µL ligation reaction was appropriate for testing of future biosensor designs.

	We have added this point to the manuscript: Lines 137-139: “Previously, we determined that 30 ng of the purified ssDNA target in the final 10 µL ligation reaction of the biosensor assay was appropriate for testing against 50 nM of each probe half¹³.”
Reviewer 3 comment 4: “Units of concentration should be standardized throughout the manuscript. In some sections, concentration is reported as µg/mL, while in others it appears simply as µg, for example.”	We thank the reviewer for their comment. For testing the biosensor designs in this current study and previous proof-of-concept paper, when testing the specificity of the probes we used synthetic ssDNA targets, such as oligonucleotides. From our previous study it was established that 50 nM of each probe half and 50 nM of each DNA target was appropriate to test the probes specificity. See Lines 123-125: “In our previous proof-of-concept study, we determined that 50 nM of each probe half and 50 nM of each 22-base long synthetic target were appropriate for testing¹³.” However, in the previous study, when testing the probes against ssDNA from the PCR amplification step of the assay we realised that real-world samples may have a range of nucleotide changes which could affect accurate molarity calculations. We therefore decided to test target amounts (ng) as a way of standardising the target samples. In our previous manuscript we found that 30 ng of target ssDNA was appropriate, and in this manuscript we mention this here: Lines 137-139: “Previously, we determined that 30 ng of the purified ssDNA target in the final 10 µL ligation reaction of the biosensor assay was appropriate for testing against 50 nM of each probe half¹³.” This is why this manuscript has both µg/mL and ng.
Reviewer 3 comment 5: “Please provide more detailed information regarding the transcription and fluorescence assay procedures.”	We thank the reviewer for their comment. We believe the required information is in the methods section: Lines 435-449: “T7 transcription and fluorescence detection: Transcription reactions (60 µl final volume) using the TranscriptAid T7 high yield transcription kit (catalogue #K0441; Thermo Scientific, Thermo Fisher Scientific, USA), comprised the following: 1.5 µl RNase Inhibitor (60 U; catalogue #M0314L; New England Biolabs), 24 µl NTP mix (final concentration of 10 mM each of ATP, CTP, GTP and UTP), 12 µl 5X TranscriptAid reaction buffer, 6 µl dithiothreitol (DTT; 0.1 M solution; catalogue #707265ML; Thermo Scientific, Thermo Fisher Scientific, USA), 6 µl TranscriptAid enzyme mix, 3 µl 2nd T7 promoter sequence oligonucleotide (1 µM stock solution), 6.3 µl of the ligation product and 1.2 µl DFHBI-1T (10 µM final concentration; catalogue #5610, Tocris Bioscience). Reactions were mixed gently via pipetting and three 15 µl aliquots of each reaction were loaded into 384-well plates (catalogue #781096, Greiner Bio-one, Kremsmünster, Austria). Reactions were measured using a CLARIOstar plate reader (BMG, Ortenberg, Germany) with the following settings: Excitation 440-15nm/Emission 510-20 nm, orbital shaking for 5 seconds at 500 rpm before measurement every 5 minutes. Delta (Δ) fluorescence measurements (a.u.) per hour were calculated between 20 and 80 minutes of the reaction runs.” Also, in the results section: Lines 92-97: “Probe A also encodes a 3’ T7 promoter while probe B also encodes the ‘Spinach’ aptamer. Therefore, after successful ligation and the addition of both a complementary T7 promoter oligonucleotide and T7 RNA polymerase, transcription of the ‘Spinach’ aptamer will occur. Subsequent binding of the transcribed aptamer with the DFHBI-1T fluorogen will then enable real-time fluorescence measurements¹³.”
Reviewer 3 comment 6: “Consider providing more detailed information about	The only statistical tests applied to the data was calculating the standard error of the mean (S.E.M). This was completed using

the statistical analyses performed. Although the software (GraphPad Prism) and the use of (s.e.m) are mentioned, it is not clear what specific statistical tests were applied to the data.”	the GraphPad Prism software as stated in the Methods: Lines 451-452: “Statistical analysis (standard error of the mean (s.e.m)) was carried out using GraphPad Prism 10.4.1 (GraphPad Software Inc., La Jolla, California).”
Reviewer 3 comment 7: “To strengthen the applicability and robustness of the developed SNAILS DNA-based biosensor, I recommend including storage stability tests. Evaluating how the sensor performs over time under various storage conditions is crucial for assessing its shelf life.”	We thank the reviewer for this comment. Whilst we agree the importance of evaluating the sensor technology over various storage conditions, such as lyophilisation of biosensor reagents/ready-to-go biosensor master mixes, we feel that this is out of the scope of this current manuscript/study. We do plan to look at storage conditions in future studies.

b

SJ_WT9	TACTATAATGTTGCGTCTAAGG
Smek_Smal_WT9	TACAATAATGATACGTTTAAAT
SM_WT9	TACGATTTT AGT CGTTAAGA
SR_WT9	TACTATTC TAAGT CGTTAAGG
SH_WT9	TACGATATAT AGT CGTGTCAAT
SB_WT9	TACAATATAGTGTGTAGAT
SG_WT9	TACGATATAT AGT TGTGTTGAT
SC_WT9a	TACAATATAT AGT CGTGTGAT
SC_WT9b	TACAATATAT CAGT CGTGTGAT

c

SJ_WT10	CACCCAGAAGTATATGTTTTGA
Smek_WT10	CATCC TGAGG TTATGTTTTAA
Smal_WT10	CATCC TGAGG TTATGTTTTGA
SM_WT10	CATCCAGAGG TT ATGTTTTGA
SR_WT10	CATCCAGAAGT TT ATGTTTTAA
SH_SC_WT10	CATCCGAGG GT ATGTTTTAA
SB_WT10a	CATCCGAGG GT ATGTTTTGA
SB_WT10b	CATCC TGAGG GTATGTTTTGA
SG_WT10	CATCCGAGG GT ATGTTTTGA

f

SJ_WT11	TCAGTTATTAGATGATCCTTATT
Smek_WT11	TCAGTTATAAGT T GATCCTTATT
Smal_WT11	TCGTTATAAGT T GGTCTTATT
SM_WT11	TCGATAAATAGTATGGGCTTATC
SR_WT11	TCGATAAATAGTGTGGGCTTATT
SH_WT11	TCATAATAATATGATCATATT
SB_WT11	TCGATAATAATCTGATCATATT
SG_WT11	TCATAATAATTTGGTCATATT
SC_WT11	TCATAATAATCTGGTCATATT

g

SJ_WT12	TATTTACTTCGGTGGTGGTATT
Smek_WT12	TTTTACTTCTAT ACT TTTGTT
Smal_WT12	TTTTACTTCTAT ACT TTTATT
SM_WT12	TATTTACGCT IGT CTTACTATT
SR_WT12	TATTTACATCTAT CT TTTATT
SH_WT12	TGTTCACTTCTAT CT TATTATT
SB_SG_SC_WT12	TATTTACTTCAATTTTATTGTT

Figure 1. Initial screening of *S. japonicum*-specific ‘SNAILS’ biosensor probes. a, Workflow of the ‘SNAILS’ biosensor protocol. **b, c, f & g,** DNA targets tested against *S. japonicum* probes 2 (b), 3 (c), 4 (f) and 5 (g). *Schistosoma* targets are indicated as follows: SJ_WT9, SJ_WT10, SJ_WT11 and SJ_WT12 (*S. japonicum*), Smek_Smal_WT9 (*S. mekongi* and *S. malayensis*), Smek_WT10, Smek_WT11 and Smek_WT12 (*S. mekongi*), Smal_WT10, Smal_WT11 and Smal_WT12 (*S. malayensis*), SM_WT9, SM_WT10, SM_WT11 and SM_WT12 (*S. mansoni*), SR_WT9, SR_WT10, SR_WT11 and SR_WT12 (*S. rodhaini*), SH_WT9, SR_WT11 and SH_WT12 (*S. haematobium*), SH_SC_WT10 (*S. haematobium* and *S. curassoni*), SB_WT9 and SB_WT11 (*S. bovis*), SB_WT10a (*S. bovis* MH647124), SB_WT10b (*S. bovis* FJ897160), SB_SG_SC_WT12 (*S. bovis*, *S. guineensis* and *S. curassoni*), SG_WT9, SG_WT10 and SG_WT11 (*S. guineensis*), SC_WT9a (*S. curassoni* AY157210), SC_WT9b (*S. curassoni* AJ519516) and SC_WT11 (*S. curassoni*). **d, e, h & i,** *S. japonicum* probe sets 2, 3, 4 and 5 respectively, were tested for their ability to recognise the corresponding 22-base *Schistosoma* target regions as indicated. No DNA corresponds to the probe negative control i.e. no target DNA. Probe sets and target concentrations were tested at 50 nM. $n = 9$ (3 replicates per reaction, each reaction split into triplicate runs). Error bars denote standard error of the mean. Probe sequences are available in Supplementary Table 1 and Δ fluorescence (a.u.) per hour values are listed in Supplementary Table 4.

Figure 2. *S. japonicum*-specific probes validation against ssDNA derived from PCR amplification of plasmid DNA. The ability of *S. japonicum*-specific probes 2, 3, 4 and 5 to bind to ssDNA derived from plasmid DNA. The 446-base *S. japonicum*-specific *cox1* target region was amplified from plasmid pAJW333 and treated as described in the methods section to produce purified ssDNA. Three PCR reactions were tested against the four probe pairs. 30 ng of target ssDNA was incubated with 50 nM of each probe half. Reactions are identified as follows: probe2_SJ1-3 (probe 2 against ssDNA samples 1-3), probe2 No DNA (probe 2 negative control), probe3_SJ1-3 (probe 3 against ssDNA samples 1-3), probe3 No DNA (probe 3 negative control), probe4_SJ1-3 (probe 4 against ssDNA samples 1-3), probe4 No DNA (probe 4 negative control), and probe5_SJ1-3 (probe 5 against ssDNA samples 1-3), probe5 No DNA (probe negative control). $n = 3$ (each reaction split into triplicate runs). Error bars denote standard error of the mean. Probe sequences are available in Supplementary Table 1 and Δ fluorescence (a.u.) per hour values are listed in Supplementary Table 5.

Figure 3. *S. japonicum*-specific probe 2 can detect *S. japonicum* cercariae samples isolated from sites in the People’s Republic of China. The ability of *S. japonicum*-specific probe 2 to recognise and bind to ssDNA derived from *S. japonicum* cercarial gDNA. The 446-base *cox1* target region was amplified from 16 different *S. japonicum* gDNA samples (SJ1-SJ16) and 3 different laboratory obtained *S. mansoni* gDNA samples (SM1-SM3) and treated as described in the methods section to produce purified ssDNA. 30 ng of target ssDNA was incubated with 50 nM of each probe half (SJ_A2 and SJ_B2). Reactions are identified as follows: SJ1-SJ16 correspond to probe 2 against *S. japonicum* ssDNA samples 1-16, SM1-SM3 corresponds to probe 2 against *S. mansoni* gDNA samples 1-3, No DNA corresponds to probe 2 negative control (no ssDNA target). $n = 3$ (each reaction split into triplicate runs). Error bars denote standard error of the mean. Probe sequences are available in Supplementary Table 1 and Δ fluorescence (a.u.) per hour values are listed in Supplementary Table 7.

Referee comment	Author response
Reviewer 2 comment 1: “Thanks authors for response to my comments.”	We thank the reviewer for their further comments.
Reviewer 2 comment 2: “However, I still think that authors need to add some clinical samples to verify the method, and it’s necessary.”	Historically there has been a significant underinvestment in diagnostics for schistosomiasis even though this neglected tropical disease causes a great burden to human and veterinary/wild animal health. As such the continued development and refinement of species-specific Schistosoma diagnostics are desirable. Alongside medical diagnostics, environmental screening of Schistosoma is an important aspect of schistosomiasis research which also feeds into public health programmes to inform their design and implementation effectiveness. For this study we tested Schistosoma japonicum cercarial samples which are the infective stage for the definitive host (Human/Water buffalo etc.). These samples were collected from a range of real-world field sites in China and were accessible to us because of our ongoing collaborations with researchers at the Natural History Museum in London. We were fortunate to be able to access difficult to attain samples and could quickly apply our biosensor technology to this species. As such we used real-world cercarial samples which were relevant to our environmental screening study aims. We acknowledge the importance of using clinical samples in situations where the study is focused on medical diagnostics and where access to these samples can be arranged. Likewise, we envision that the ‘SNAILS’ biosensor technology can potentially be applied to clinical samples in future studies. However, we currently have no access to human clinical samples which would be a huge undertaking and so are currently unable to test these types of samples. However, given our environmental screening focus and our use of real-world field samples that are relevant to our studies stated purpose, we respectfully disagree that clinical samples are essential to the current study. To further clarify these aspects, we have also amended the manuscript with the following: lines: 286-302: “Collectively, these two studies now demonstrate that ‘SNAILS’ biosensors can be used to detect for the three most prevalent Schistosoma species that infect humans (S. mansoni, S. haematobium and S. japonicum). However, the current studies shift in focus towards S. japonicum necessitated meaningfully different approaches than our previous study, including the collection, sequencing and bioinformatic analyses of new S. japonicum samples from appropriate geographical regions (PRC, the Philippines, Japan and Taiwan) which were beyond the scope of our previous Africa-focused schistosomiasis biosensor project. These data also informed the design and validation of several new ‘SNAILS’ biosensor probes for the species-specific detection of S. japonicum. Likewise, ‘SNAILS’ biosensor workflows also had to be adapted beyond the use of adult worm gDNA samples to now also enable the utilisation of cercariae samples taken from real-world field sites in PRC. The testing of these new samples also further demonstrates the potential utility of the ‘SNAILS’ biosensor for use in future schistosomiasis environmental DNA screening studies. Whilst we envision that ‘SNAILS’ biosensor workflows could potentially be adapted to

	also incorporate testing of human faeces or urine samples, these types of samples were not available to, nor were they directly relevant to, the present study. This is primarily due to our studies core focus on environmental sampling for schistosomiasis research applications.”
Reviewer 2 comment 3: “Authors can use some other certain organisms’ genomic DNA to verify the specificity of the method. In clinical samples, these genomic DNA of some uncertain organisms are complex. Thus, authors could not to make sure that the specificity of the method are excellent. Authors also mentioned that ‘this method can detect and differentiate small changes in the short target sequence’. This also means if some small changes in the target sequence, the sample might be missing out. How did authors to make sure the conservation of these targets in S. japonicum from all epidemic areas? Such as the Philippines strain or some strains from other southeast Asian countries.”	We further thank the reviewer for highlighting this important point about different geographical strains of S. japonicum i.e. those found in China versus those isolated from the Philippines and other Southeast Asia countries. For this study, when we further interrogated the potential probe targets we previously identified for S. japonicum, the available cox1 FASTA sequences we used were isolated and obtained from different geographical locations as indicated in Supplementary Table 2 (both China and the Philippines). However, in response to suggestions from a previous reviewer, we further interrogated GenBank and found further cox1 FASTA sequences for S. japonicum isolated in Japan and Taiwan (see Supplementary Table 2). We have also repeated the sequence alignment using MUSCLE and the updated set of cox1 FASTA sequences and have updated Supplementary Figure 1 accordingly. The four probe targets we identified are present in all of the cox1 FASTA sequences we analysed across several different geographical locations. This, therefore, indicates that the probes in this study are suitably designed to account for geographical variation in strains and have shown that they can detect S. japonicum sequences isolated from several different geographical regions (China, the Philippines, Japan and Taiwan). To further clarify this, we have edited the manuscript as follows: Lines 110-113: “These included sequences obtained from a range of locations in PRC, the Philippines, Japan and Taiwan, thereby enabling the identification of S. japonicum targets covering a range of geographical strains/isolates of this species. Four of these targets were selected for further molecular analysis.” The method section was also changed as follows: Lines 333-337: “A second MUSCLE¹⁶ alignment, using default parameters, of EU325878 and other unpublished and published (KR855674⁴⁸ and EF635954⁵²) S. japonicum cox1 nucleotide sequences obtained from GenBank was undertaken, to further validate the S. japonicum-specific cox1 targets identified (Supplementary Figure 1).” Regarding the point about the ‘SNAILS’ biosensor being able to detect and differentiate small changes in the target sequence, we see that as a benefit. This enables for the detection of species with minimal differences in the target gene(s). The biosensor technology can also be evolved quickly. Indeed, the present study further demonstrates that new SNAILS probe designs can be iterated and produced quickly, to react to, if necessary to new sequences when they become available. Noting also, that even when probe designs are changed the overall workflows remains standardised.
Reviewer 2 comment 4: In authors’ previous study, they tested specificity of the method for S. mansoni. In this study, authors used S. japonicum as the research object, it also needed to retest the specificity of these targets of S. japonicum	In our previous biosensor study, we aligned our research according to the needs and priorities of the Africa-focused Water Infrastructure for Schistosomiasis-Endemic Regions (WISER) project. WISER was focused on S. mansoni and S. haematobium detection and mitigation strategies as these are the main human infecting species present in Africa. Therefore, as part of our

distinguishing from some other organisms. Meanwhile, authors also provided the data that the detection limit of genomic DNA of *S. japonicum*. Did authors only change the research object (*S. japonicum*) in this study compared to their previous study (*S. mansoni*)? Why did authors not use the method in the previous study to detect clinical samples? Such as water, feces or urine.

previous study, we tested the biosensor against genomic DNA (gDNA) isolated from adult worm samples provided by the Natural History Museum, London.

In contrast, for the present study we expanded upon our recently developed 'SNAILS' biosensor technology to detect and differentiate between *S. mansoni*, *S. haematobium* and now also with a new key focus on *S. japonicum*. These three species are the most prevalent agents of human schistosomiasis. For this study new probes were also designed and synthesised enabling the detection of *S. japonicum* isolated from different geographical regions (China, the Philippines, Japan and Taiwan) which are new to this study. We then tested the probes against the life cycle stage that infects the definitive host (humans etc.), namely cercariae, collected from different sites in China. Consequently, we believe that our shift in this study to focus towards *S. japonicum*, our development of new probes, new bioinformatic analysis of *S. japonicum* sequences and testing of different kinds of samples that are also from new geographical locations is a significant shift from our previous study. These new data also help to expand upon and further demonstrate the capabilities of the SNAILS biosensor technology.

We, as advised by the Editors, have also highlighted the differences between the two studies in the discussion section, lines: 286-298: "Collectively, these two studies now demonstrate that 'SNAILS' biosensors can be used to detect for the three most prevalent *Schistosoma* species that infect humans (*S. mansoni*, *S. haematobium* and *S. japonicum*). However, the current studies shift in focus towards *S. japonicum* necessitated meaningfully different approaches than our previous study, including the collection, sequencing and bioinformatic analyses of new *S. japonicum* samples from appropriate geographical regions (PRC, the Philippines, Japan and Taiwan) which were beyond the scope of our previous Africa-focused schistosomiasis biosensor project. These data also informed the design and validation of several new 'SNAILS' biosensor probes for the species-specific detection of *S. japonicum*. Likewise, 'SNAILS' biosensor workflows also had to be adapted beyond the use of adult worm gDNA samples to now also enable the utilisation of cercariae samples taken from real-world field sites in PRC. The testing of these new samples also further demonstrates the potential utility of the 'SNAILS' biosensor for use in future schistosomiasis environmental DNA screening studies."

We did not test clinical samples such as faeces or urine as we did not and still do not have access to these kinds of samples. As noted above our study was focused more on environmental screening and to this end we tested real world samples from different geographical locations.

To further clarify these aspects, we have also amended the manuscript with the following lines: 298-302: "Whilst we envision that 'SNAILS' biosensor workflows could potentially be adapted to also incorporate testing of human faeces or urine samples, these types of samples were not available to, nor were they directly relevant to, the present study. This is primarily due to our studies core focus on environmental sampling for schistosomiasis research applications."